# Plant Transglutaminases: New Insights in Biochemistry, Genetics, and Physiology

**DOI:** 10.3390/cells11091529

**Published:** 2022-05-03

**Authors:** Luigi Parrotta, Umesh Kumar Tanwar, Iris Aloisi, Ewa Sobieszczuk-Nowicka, Magdalena Arasimowicz-Jelonek, Stefano Del Duca

**Affiliations:** 1Department of Biological, Geological and Environmental Sciences, University of Bologna, Via Irnerio 42, 40126 Bologna, Italy; luigi.parrotta@unibo.it (L.P.); iris.aloisi2@unibo.it (I.A.); 2Interdepartmental Centre for Agri-Food Industrial Research, University of Bologna, Via Quinto Bucci 336, 47521 Cesena, Italy; 3Department of Plant Physiology, Faculty of Biology, Adam Mickiewicz University in Poznań, Uniwersytetu Poznańskiego 6, 61-614 Poznań, Poland; umetan@amu.edu.pl (U.K.T.); ewa.sobieszczuk-nowicka@amu.edu.pl (E.S.-N.); 4Department of Plant Ecophysiology, Faculty of Biology, Adam Mickiewicz University in Poznań, Uniwersytetu Poznańskiego 6, 61-614 Poznań, Poland; magdalena.arasimowicz@amu.edu.pl

**Keywords:** plant transglutaminase, bioinformatics, biochemical features, physiological roles

## Abstract

Transglutaminases (TGases) are calcium-dependent enzymes that catalyse an acyl-transfer reaction between primary amino groups and protein-bound Gln residues. They are widely distributed in nature, being found in vertebrates, invertebrates, microorganisms, and plants. TGases and their functionality have been less studied in plants than humans and animals. TGases are distributed in all plant organs, such as leaves, tubers, roots, flowers, buds, pollen, and various cell compartments, including chloroplasts, the cytoplasm, and the cell wall. Recent molecular, physiological, and biochemical evidence pointing to the role of TGases in plant biology and the mechanisms in which they are involved allows us to consider their role in processes such as photosynthesis, plant fertilisation, responses to biotic and abiotic stresses, and leaf senescence. In the present paper, an in-depth description of the biochemical characteristics and a bioinformatics comparison of plant TGases is provided. We also present the phylogenetic relationship, gene structure, and sequence alignment of TGase proteins in various plant species, not described elsewhere. Currently, our knowledge of these proteins in plants is still insufficient. Further research with the aim of identifying and describing the regulatory components of these enzymes and the processes regulated by them is needed.

## 1. Introduction

Transglutaminases (TGases) (EC 2.3.2.13) are protein–glutamine γ-glutamyl transferases that are calcium (Ca^2+^)-dependent in all organisms, except bacteria. The term TGase was first introduced in 1959 [1], when an enzyme showing transamidating properties in a guinea-pig liver was first discovered [2]. TGases catalyse the acyl-transfer reaction between one or two primary amino groups, such as those of polyamines (PAs), to the γ-carboxyamide group of protein endo-glutamine residues [3].

TGases are known to be widely distributed in nature, being found in vertebrates, invertebrates, molluscs, plants, and microorganisms [2,4]. Among plants, TGase activity has been reported in angiosperms [5,6] and studied in several cellular processes. It is distributed in different organs, such as leaves, tubers, roots, flowers, buds, and pollen, as well as in various cell compartments, including chloroplasts, the cytoplasm, and the cell wall [7,8]. TGases have been reported as being associated with growth (e.g., cell cycle, apical growth, seedling growth, and root growth), pollen–pistil interactions, differentiation, programmed cell death, and stress responses [9,10,11,12,13], as well as being involved in maintaining the stability of photosynthetic membrane proteins via the covalent binding of PAs to membrane protein complexes [14].

In this review, we describe the biochemical characteristics and provide a bioinformatics comparison of plant TGases and data on the physiological roles proposed for these enzymes by referring to the literature of the last 10 years. Specifically, the review is focused on the involvement of TGases in photosynthesis, fertilisation, biotic and abiotic stress responses, and leaf senescence.

## 2. Biochemical Features of TGases

TGases are able to catalyse three reactions: (i) deamidating glutamine (Gln) residues, (ii) transamidating activity with amine incorporation into protein endo-Gln residues, and (iii) crosslinking between Gln and Lys protein residues [15]. Concerning the last reaction, TGases catalyse crosslink reactions between two proteins or two residues of the same protein, always involving Lys and Gln residues [5]. The inter- and intra-protein X-links catalysed by TGases lead to structural and chemical modifications that ultimately affect the functional properties of these proteins or allow for further structural forms and the creation of polymers (in the case of X-links between more proteins) of high molecular weight. 

TGases and their functions have been less studied in plants than in humans and animals, mostly mammals [16]. Present knowledge about the biochemical features of plant TGases is fragmentary, as different studies have focused on different aspects of TGase biochemistry. To better clarify the biochemical properties of plant TGases, we summarise the main characteristics of these enzymes in this paper. 

The first evidence for the presence of TGases in plants was reported in 1987 [6,17]. More than 20 years later, Suzuki and co-workers found that the core catalytic domain of a peptide, N-glycanase (PNGase), contains the Cys, His, and Asp catalytic triads, which are highly conserved in eukaryotic TGases [18]. Based on the homology of the TGase domain, the first plant TGase was identified in *Arabidopsis thaliana* (AtPNG1). In *Arabidopsis*, a single gene, encoding a putative N-glycanase and containing the Cys–His–Asp catalytic triad, was identified. AtPNG1 consists of 721 amino acids and has a molecular weight of 86 kDa. In particular, the triad domain in AtPNG1 is Cys251–His278–Asp295 [19]. These observations were confirmed in *Oryza sativa*, where OsPNG1, a protein with 447 amino acids, has shown 62% homology and 75% similarity with AtPNG1, which is covered by 427 amino acids in this case [20]. In *Cucumber sativum*, the *CsPNG1* gene consists of a 1836-bp open reading frame encoding a protein of 611 amino acids. In soybean (*Glycine max*), GmPNG1 was predicted to encode for 720 amino acids with 54% similarity to the amino acid sequence of CsPNG1. Furthermore, the conserved domain near the catalytic triad has shown a similarity of 72% with AtPNG1 [21]. 

Initially, research on TGases mainly focused on their biochemical inhibition or activation. Early studies on plant TGases confirmed their Ca^2+^-dependency and inhibition by different chemicals, such as dithiothreitol (DTT), copper (Cu^2+^), ethylenediaminetetraacetic acid (EDTA), ethyleneglycoltetraacetic acid (EGTA), and o-phenanthroline (a metalloprotease inhibitor of TGase activity). All of these inhibited and/or decreased the activity of plant TGases in vitro [6,17]. The Ca^2+^-dependency of plant TGases was confirmed in numerous papers, but the first clear evidence was reported by Bonner’s group [22]. Del Duca and co-authors reported that the plastidial TGase was not only Ca^2+^-dependent but also light-stimulated and that it catalysed the incorporation of PAs into thylakoid and stromal proteins in *Helianthus tuberosus* [23]. TGase Ca^2+^-dependency was also confirmed in the roots and leaves of *Pisum sativum*, *Phaseolus vulgaris*, wheat, and barley, where the enzyme’s activity was inhibited up to 80% in pea roots and leaves and 100% in all other cases analysed with 1 mM EDTA and/or EGTA [22]. TGase activity decreased by 34% in pea roots and 24% in pea leaves following the addition of 10 mM iodoacetamide. This inhibition was not reported with 1 mM guanosine triphosphate (GTP) and at a low Ca^2+^ concentration (1 μm free Ca^2+^), indicating that, in this respect, plant TGases may be different from mammalian ones and suggesting that the active sites of plant TGases may be similar but not identical to the active sites of mammalian tissue enzymes. Furthermore, the activity of mammalian tissue TGases is regulated by GTP at low concentrations of Ca^2+^ [22]. In *Arabidopsis*, AtPNG1p was demonstrated to efficiently polymerise bovine serum albumin (BSA) in a Ca^2+^- and DTT-dependent manner. Both GTP and EGTA inhibited enzyme activity, thought it was not affected by magnesium, sodium, or potassium [19]. The TGase activity in chloroplasts isolated from the leaves of *Helianthus tuberosus* was inhibited by SH reagents, such as dithiobis-ethylamine (DTEA), N-ethylmaleimide (NEM), and DTT. In particular, DTT showed an inhibitory effect that reached its maximum at 10 mM (98% inhibition at 10 mM and 34% at 1 mM), whereas 1 mM DTEA caused 86% inhibition and 10 mM NEM caused 43% inhibition [24]. In lupine seedlings, DTT slowed down the rate of casein polymerisation induced by TGase activity [25]. Del Duca and co-workers reported that the in vitro TGase activity of chloroplasts was enhanced by 1 mM Ca^2+^ and severely inhibited by 1 mM EGTA in a dose-dependent manner [24]. Additionally in *Oryza sativa*, TGase activity was shown to be increased by exogenous Ca^2+^, and inhibited by EGTA. The presence of specific compounds, such as GTP, monodansyl cadaverine (MDC), and DTT, completely inhibited TGase activity [26]. In the green alga *Chlamydomonas reinhardtii*, TGase activity was not impaired by of 1 mM Zn^2+^ but was completely blocked by NEM and p-chloromercuribenzoate [27]. 

The TGase of maize leaf, when expressed in bacteria, was found to be inhibited by the competitive substrate MDC, GTP, and in the absence of exogenous Ca^2+^. TGase activity in chloroplasts (thylakoids and grana) was inhibited by MDC, GTP, diethyldithiocarbamic acid (DIECA), and 3-(3,4-dichlorophenyl)-1,1-dimethylurea (Diuron) [28]. In *Rosmarinus officinalis*, TGase activity was increasingly stimulated by 2–6 mM CaCl_2_ (from 5 to 20%), and was not inhibited by 2–14% NaCl [29]. In addition, TGase activity is affected by different ions; in particular, it was reported that Mg^2+^ had a slightly inhibitory effect [24]. Moderate inhibition was reported for Fe, Cu, and Mn; differently, in rosemary, the TGase was not inhibited by Mg, Ba, and Zn [29]. Shu and co-authors [30] reported an upregulated activity in the presence of NaCl, but in the presence of o-phenanthroline, the gene expression level of TGase declined and the application of exogenous o-phenanthroline significantly decreased endogenous PAs content in cucumber leaves.

The inhibitory roles and precise functions of proteases are still unclear. What is known is that proteases can cleave TGase substrates, thereby favouring accessibility to the binding sites. Furthermore, in mammalian cells and microorganisms, TGases can be directly activated by protease-induced processing [31,32], and this activation mechanism cannot be excluded in plants. It has been hypothesised that the direct action of protease inhibitors may inactivate the Cys thiol group in the active site of plant TGases [24].

The influence of biogenic diamines on enzyme activity has also been exhaustively studied. When TGase activity was checked by testing the incorporation of ^3^[H]putrescine (Put) into N-N′-dimethyl casein, cadaverine showed a higher apparent inhibition of TGases than diaminopropane [6]. Spermidine (Spd) and spermine (Spm) showed a better incorporation than Put in sprout apices of *H. tuberosus* and in apical meristematic tissue of etiolated pea seedlings. Additionally, 5 mM histamine, a competitive substrate of TGases, caused a 64% inhibition of the activity, as measured by the incorporation of labelled PAs into protein substrates [17]. Several authors have reported that TGase activity was affected by different substrates and amine concentrations; in particular, the large subunit of RuBisCO was shown to be a protein substrate in *Medicago sativa* [33,34].

TGase activity in plants impacts the photosynthetic machinery. The enzyme was shown to be light-inducible in *Quercus ilex* [35]. Likewise in rice, TGase activity was shown to be light-sensitive and completely inhibited by darkness [36]. A recent study indicated that the overexpression of TGases could promote the CO_2_ assimilation rate by activating Calvin cycle enzymes [37].

Only few researchers have attempted to highlight the enzymatic kinetics of plant TGases. Michaelis–Menten kinetics were calculated by testing the incorporation of ^3^[H]Put into N-N′-dimethyl casein catalysed by a pea seedling TGase. The Lineweaver–Burk plot of the data showed an apparent V_Max_ of 41 nmol/mg protein h and an apparent K_M_ of 9.63 mM of Put [6]. A purified recombinant maize TGase had a K_M_ of 3.98 µmol L^−1^ and a V_Max_ of 2711 µmol L^−1^ min^−1^, as calculated with a fluorometric method [38].

The presence and distribution of TGases in some angiosperms and algae with their main biochemical features is presented in Table 1. Plant TGases have been identified in different cell compartments. Molecular weight varies significantly, from the 30 kDa thylakoid-localised TGase isolated from *Cucumber sativus* cotyledons [39] to the 150 kDa TGases found in *Zea mays* thylakoids and grana extracts [28] and the 160–180 kDa band found in mature maize pollen [40]. However, the most frequently found form, based only on molecular weight, is the 58 kDa one. Most reports have indicated that the optimum pH for TGase activity assays falls within the range of 7.5–8.5.

## 3. Bioinformatics Analyses

Bioinformatics analyses, such as comparisons of gene sequences, can support biochemical data and add new knowledge regarding the phylogenetic relationships and genomic organisation of TGases in plants. Here, we present a comparison by sequence alignments, phylogenetic relationships, and data on the genomic organisation of different TGases in various plant species for the first time. 

To date, TGases have been identified in an increasing number of plant species, but a comparative analysis of their characteristics has not been performed. We selected the TGase family members from model plants [57] available in the PLAZA_5.0 database (https://bioinformatics.psb.ugent.be/plaza/, accessed on 17 March 2022): a total of 41 TGase genes were found to be distributed in 30 plant species (Figure 1). Among angiosperms, *Glycine max*, *Nicotiana tabacum*, and *Miscanthus sinensis* have two duplicated genes each and *Eucalyptus grandis* has three duplicated genes. Notably, *Selaginella moellendorffii* (Lycophyta) has five duplicated genes. Thus, gene duplication seems to have played a dominant role in the expansion of the TGase family in plants. Gene duplication, expansion, and subsequent diversification are features of the evolutionary process. The abundance of duplicate genes in plant genomes originated from ancient duplication events and a high rate of the retention of extant pairs of duplicate genes. These duplicates have contributed to the evolution of novel functions, such as in growth and development, disease resistance, and stress tolerance [58].

The phylogenetic analysis (Figure 1) classified the plant TGases into taxonomic groups, i.e., monocots, dicots, bryophytes, lycophytes, marchantiophytes, and chlorophytes. The monocots and dicots (angiosperms) form a separate clade, suggesting that they are more evolutionarily divergent than the other species. This analysis is consistent with plant evolutionary history. The gene structure analysis (Figure 1) showed that angiosperms have different intron/exon arrangements compared to other plant taxa, though no major differences were observed between monocots and dicots. TGase gene size varied from 2 to 28 kbp in most of the examined plants; however, it was significantly larger in *Vitis vinifera* (54 kbp) and barley (60 kbp). Though *V. vinifera* was found to have genome size of only ~500 Mb, its TGase gene was large with long introns. This might be because of the repetitive/transposable elements (TEs) abundance (41%) in the grapevine genome [59]; moreover, introns are quite rich in repeats and TEs. The large size of the barley TGase might also have been due to the specific characteristics of its genome, which is rich in pseudogenes and small gene fragments mainly located towards chromosome tips or as tandemly repeated units [60]. These repetitive regions are present in introns and/or intergenic spaces [61,62]. Most plant TGases include one conserved large exon, which might be associated with the enzyme’s active site. The level of conservation of plant TGases was compared to animal and microbial ones via ConSurf analysis using PF01841 (Transglut_core, https://pfam.xfam.org/family/PF01841 accessed on 17 March 2022).

The multiple sequence alignments (Figure 2) showed that TGases had a highly conserved domain typical of these enzymes, which consists of the Cys–His–Asp catalytic triad. The protein sequence of Misin04G291400 (*Miscanthus sinensis*) was found to differ in the catalytic residues; this might have been due to the partial sequence availability. Plant TGase protein sequences are less conserved than those of animals and microbes at the N- and/or C-terminus; however, this does not affect the catalytic activity. Plant TGases without conserved catalytic domains might have arisen due to a unique deletion/substitution in the genome. For example, despite high similarity at the nucleotide level, the unique deletion of guanine (G) in the maize TGZ15/TGZ21 cDNA sequences resulted in a frameshift in their amino acid sequences and, consequently, a lack of homology at their C-termini, where the TGase catalytic triad was found by Villalobos et al. [28]. A rice homolog of the TGZ15/TGZ21 proteins, named TGO, also possesses a unique G deletion [26] that is not observed in other available rice sequences, including the genomic sequence of rice chromosome 4. The maize TGZ15/TGZ21 and rice TGO proteins share a high sequence similarity (70%), though only upstream of the G deletion positions, while their C-termini differ in length, sequence, and catalytic triad localization. However, overexpression experiments have provided evidence that both proteins possess TGase activity [7,14,26,28,63,64,65], although their catalytic domains are not conserved. Overall, the bioinformatics analysis suggests that most plant TGase sequences are conserved in nature. Although a few exhibit quite different features, they still exert a similar function.

## 4. Physiological Role of TGases in Plants

In plants, TGases have been primarily studied by focusing on the molecular mechanisms linking PAs to proteins by inter- and intramolecular bonds. These findings have been correlated to several aspects of growth and differentiation, as well as to stress responses [5,15]. Research on plant TGases has been hampered by difficulties in the purification of the enzyme and by the limited/scarce sequence identities between animal TGases and those reported in the available plant databases [66]. In general, studies on plant TGases have mainly dealt with its distribution and function [48].

It is known that TGases are present in most plant organs and organelles. Here, we report the most recent evidence for their involvement in various processes, since the last extensive review on this topic did not account for the last decade of results [5,66]. A schematic model of the main physiological roles of plant TGases is shown in Figure 3.

### 4.1. TGases and Photosynthesis

Villalobos et al. first reported that a TGase was mainly present in the grana-appressed thylakoids of light-exposed maize chloroplasts [54]. The activity of maize TGase was found to be inhibited by GTP, DTT, and other compounds but significantly increased when the enzymatic assay was performed in the presence of light [49]. The gene sequence analysis showed that maize TGase possesses a chloroplast import peptide composed of 47 amino acids and B-type repeats that are located in a non-catalytic domain of the enzyme. The overexpression of the maize plastidial TGase increased the activity of TGases in thylakoids of *Arabidopsis thaliana* [67]. In chloroplasts, TGases appear to stabilise both the photosynthetic complexes and Rubisco. Being regulated by light and other external factors, TGases might exert a photoprotective effect on photosynthesis [7]. The overexpression of TGases has been shown to increase the CO_2_ assimilation rate through the activation of Calvin cycle enzymes in tomato leaves [37]. Changes in cellular redox homeostasis have been proposed to be involved in the activation of Calvin cycle enzymes [37]. The enhanced TGase-mediated binding of PAs to thylakoid membranes is surely involved in the aggregation of the light-harvesting complex (LHCII), which exerts a key regulatory role in dissipating excess excitation energy, thus improving photochemical efficiency under salt stress [68]. TGase activity increases salt stress tolerance in cucumber plants due to an increased endogenous PAs content and ROS scavenging capacity, as well as the promotion of carbon assimilation and photosynthetic products. However, the mechanism by which TGase regulates the photochemical efficiency of plants under salt stress remains unclear [30]. Plastidial proteins involved in photoprotection and in promoting the thylakoid electrochemical gradient are TGase substrates. Consequently, TGase interconnects more PSII proteins with other photoprotective and proton motive force (PMF) proteins (e.g., LHCII and ATPase); moreover, TGase changes the balance of pmf, thereby increasing the PA-linked protein pool [67]. A recent study confirmed that a *PNG1* gene containing a typical TGase catalytic triad domain, like that of *AtPNG1*, plays a positive role in improving plant salt tolerance in cucumber plants [21].

### 4.2. TGases and Plant Fertilisation

Several studies have highlighted the involvement of TGases in the fertilization process of angiosperms. In particular, the enzyme plays a role in pollen–pistil recognition and pollen rejection; it is a crucial factor for pollen tube growth, being involved in the organisation of cytoskeleton proteins [49,69]. Moreover, several plant models suggest that TGase activity is involved in the self-incompatibility response [9,12,13,70].

TGases have been reported to not only be localised inside the pollen tube but also exist extracellularly. In the pollen tube cytosol, TGases modify cytoskeletal proteins, thereby regulating apical growth. Some reports also suggest an extracellular localisation of the enzyme and its involvement in pollen tube cell-wall construction and organisation [5,49,71]. During pistil fertilisation, the pollen tube grows through the stigma and style following a precise set of extracellular signals including PAs, which can regulate the growth of the pollen tube [72,73]. In fact, in in vitro pollen germination experiments, PAs were released into the germination medium together with other factors (RNAs and proteins). It has been reported that an extracellular TGase is required for apple pollen tube growth, suggesting its possible involvement in pollen tube and style adhesion, thus favouring cross-talk between male and female counterparts [74]. In addition to a cytosolic form of TGase, data suggest the existence of TGase forms associated with the internal membranes and the cell wall of pollen tubes. This different localization extends the functional range of pollen TGases that can be precisely redistributed in different cellular compartments [75]. The presence of an extracellular TGase raises a question regarding the function locally exerted by this enzyme.

### 4.3. TGases and Biotic Stress Responses

TGases may play an important role in plant–pathogen interactions and the resulting defence responses. Thus, TGases are involved in the hypersensitive reaction (HR), which consists of programmed cell death at the site of pathogen entry and it is associated with restriction of pathogen multiplication and spread [76,77]. The HR is accompanied by an increase in TGase activity and its products that are distributed in different fractions, though mainly in those containing proteins released from membranes and cell walls by high ionic strength and detergents [11]. The synthesis of mono-(γ-glutamyl)-Put and bis-(γ-glutamyl)-Spd, which represent solid evidence of TGase catalysis, revealed that both transamidating and cross-linking activity were enhanced in leaves undergoing the HR but not in mock controls [78]. In a recent article, healthy susceptible and resistant to *Phytophthora capsici* pepper (*Capsicum annuum*) plants showed a very similar pattern of TGase accumulation, thought it was distinct when inoculated with the pathogen. Such differently expressed post-infection patterns of TGases indicate that the defence mechanism of resistant plants might be based on the activation of specific plant TGase isoforms. These data support the hypothesis that TGases play role sin defence responses against some pathogens [42], such as *Phytophthora* spp., one of the most dangerous plant pathogens.

TGase activity has also been detected in different *Phytophthora* species. A 42-kDa cell wall-associated TGase (GP42) of *Phytophthora sojae* was found to contain a surface-exposed fragment called PEP-13 that acts as an elicitor of defence responses in parsley and potato [79,80]. The PEP-13 motif was reported to be highly conserved in several *Phytophthora* species. TGases activate defence responses, thus suggesting their function as genus-specific recognition determinants in host and non-host plants [15]. In addition, TGase structural sequences with eliciting activity, associated with plant defence mechanisms, were isolated and characterised in *Phytophthora cinnamomi*. The fragments were found to encode a 533 deduced amino acid protein that includes an ORF with high identity similarity to *Phytophthora sojae* (70%), *Phytophthora megasperma* (70%), and *Phytophthora infestans* (61%) TGases. The alignment of a TGase gene with several TGase proteins revealed that the protein contains the conserved catalytic domain [81]. In addition, a recent study on the biochemical characterization of an acyltransferase enzyme, responsible for the pathogenicity of *Phytophthora melonis*, indicated that this protein possesses two domains, A (ranging from residues 260 to 620) and B (ranging from 141 to 219). The A domain possesses TGase-elicitor properties [82].

### 4.4. TGases and Abiotic Stress Responses

TGases regulate the posttranslational modification of proteins involved in a wide range of plant responses to environmental stresses. In general, the stress-related function of TGases could be ascribed to a positive relationship between enzyme activity, PA biosynthesis, and the photosynthetic efficiency maintained by the activation states of the Calvin cycle [37]. In addition, TGase-improved photosynthetic capacity seems to be supported by changes in the cellular redox status and activation of antioxidant enzymes [37]. This was confirmed by studies showing that TGase-deficient mutants (*tgase-1* and *tgase-2*) of tomatoes exhibited a decreased activity of antioxidant enzymes engaged in the ascorbate (AsA)–GSH cycle, while *TGase*-overexpressing (*TGase*OE) plants showed enhanced activities of ascorbate peroxidase (APX), dehydroascorbate reductase (DHAR), and glutathione reductase (GR). High TGase activity in *TGase*OE plants also correlated with significantly increased ratios of both GSH/GSSG and AsA/DHA [37]. This upregulated antioxidant machinery can prevent redox homeostasis misbalance provoked by the over-accumulation of reactive oxygen and nitrogen species under stress conditions. More recently, Jahan et al. [83] reported that the overexpression of TGases in tomatoes enhanced tolerance to heat stress. A comparative transcriptomic study between wild-type (WT) and *TGase*OE tomato plants revealed that in TGase-induced heat tolerance, a crucial role is played by genes associated with pathways responsible for protein processing in the endoplasmic reticulum, as well as carbon fixation [83]. Moreover, the specific high-temperature response of *TGase*OE plants was associated with increased expression of heat-induced heat shock factors compared to WT plants, which was consistent with the expression patterns of heat shock proteins, thus indicating that heat shock factors might perform a pivotal role in the thermotolerance of *TGase*OE plants.

An enhanced salt tolerance was observed in tobacco plants overexpressing cucumber *CsTGase* [84]. The transgenic plants showed vigorous growth and higher net photosynthetic rate, as well as stomatal conductance. In turn, the *CsTGase*-induced salt tolerance was associated with increased levels of chloroplast PAs, enhanced transcript levels of photosynthesis-related genes, and the accumulation of thylakoid membrane proteins such as D1 and D2 [84]. It is noteworthy that significantly higher TGase activity was observed in a salt-tolerant cultivar of cucumber in comparison to a salt-sensitive one [30,85]. The TGase-mediated tolerance to NaCl was proven by spraying leaves with 1 mM o-phenanthroline, which resulted in decreased bound PA levels, the decreased photochemical efficiency of PSII, and the growth reduction of both cucumber cultivars [30].

Interestingly, TGases appear to play functional roles during acclimation to high salinity levels. As indicated in the green halophilic microalga *Dunaliella salina,* acute hyper-saline stress under light caused an immediate change in the concentration of chloroplast TGases with concomitant variations in enzymatic activity [44]. Moreover, a PA-deficient variant of *Dunaliella* exhibiting low TGase activity was found to be more severely affected by salt stress; however, put application visibly recovered TGase activity and led to considerable enhancements of chlorophyll *a* and *b* content [44]. In the marine macroalga *Grateloupia doryphora* (Mont.) Howe exposed to moderate hyposaline conditions, diminished TGase activity correlated with an increased pool of free PAs [86]. The positive relationship between TGases and free PAs could constitute a simple metabolic adjustment during acclimation to hyposaline conditions, since free PAs were found to be able to increase photosynthetic rate in the macroalga.

According to Pinto-Marijuan et al. [35], TGases are involved in adaptations to different light conditions. Holm oak leaves exposed to darkness until midday and then subjected to abrupt high light intensity showed enhanced TGase activity, resulting in the maximum accumulation of bound Put. The photoprotective role of TGases was hypothesised to be due to their enhanced activity during increasing light intensity, as previously observed in the systematically distant PA-deficient strain of *Dunaliella* [44]. Although TGase activity in the microalga was induced by salt stress, it was always higher in the light than in the dark [5].

Finally, TGases could also mediate the response to wounding and the wound-healing process. As documented by Serafini-Fracassini et al. [87], TGase activity was enhanced in tuber explants of *Helianthus tuberosus* as a result of wounding, in which the enzyme triggers the resumption of the cell cycle. This highlights the role of TGases in linking abiotic and biotic stimuli since insect/herbivore feeding and pathogen attack is often related to plant tissue injury. Recently, the involvement of TGases in wounding was also reported in *Arabidopsis thaliana* [88]. In this experimental model, an *Atpng1* knockout (KO) line was analysed during plant development and under heat and wounding stress. WT and KO lines were compared in terms of response to wounding and recovery from wounding (e.g., the formation of a scarring tissue that covered the entire wound). TGases accumulated differently in the two lines: in the stem of the WT line, TGases were mainly localised in the 2–3 cell layers underneath dead cells on the stem surface, thus suggesting their involvement in the wound healing process (probably by exerting a gluing function and by strengthening cell walls), as previously observed during senescence in *Nicotiana tabacum* petals [51]. In the KO line, the lack of TGase activity in the cell walls may have been related to the observed weaker anatomical structure, characterised by parenchyma with large spherical cells and wide intercellular spaces. These features were probably due to the reduced stiffness of the KO cell walls, failing to counteract the internal turgor pressure [89]. In WT leaves, a rapid increase in TGase activity was observed; within 15 min, it was about three-fold higher than the basal activity observed at time 0 and then decreased. On the contrary, in KO leaves, the wound was effective, as activity remained at a constantly low level for 24 h. In the WT line, wounding-induced *AtPNG1* transcript accumulation was observed within the first 5 min and reached minimum levels after 15–30 min. The potential involvement of TGases in the healing processes is still poorly understood for plants, but it has already been demonstrated for animal tissues [90,91]. These preliminary results suggest that the enzyme plays a role in plant wounding responses.

### 4.5. TGases and Leaf Senescence

In plants, senescence is a highly controlled and active process requiring global metabolic reprogramming, aimed at the organised disintegration and remobilization of valuable resources. It is a fundamental aspect of plant development that is necessary to optimise resource allocation and promote phenotypic plasticity in order to acclimate to adverse environmental conditions [92]. Structural changes of the chloroplast, eventually resulting in chloroplast degradation, mark the first phase of a sequential process that leads to leaf senescence, both developmental and stress-induced [93]. 

Physiological and structural changes in chloroplasts during senescence are associated with PA conjugation, modifications of chloroplast proteins, and the modulation of chloroplast-localised TGases (ChlTGases). The barley ChlTGase was found to be activated during dark-induced leaf senescence, which is associated with enhanced local TGase accumulation and activity, as well as the increased expression of the barley *HvPng1*-like gene [3]. Results with barley leaves also showed that TGase activity was lower when the samples were incubated with cytokinin, a phytohormone known for its anti-senescence properties [3]. The ChlTGase localization within chloroplast structures, as well as the identification of the post-translational modification of plastid proteins (PA-conjugated proteins), suggested a notable contribution of ChlTGases to the dark-induced senescence-associated process, including stress response, photosynthesis inhibition, and cell death manifested by the chloroplast-to-gerontoplast conversion and subsequent degradation [94]. In situ localization and changes in ChlTGase activity during dark-induced senescence were shown to mirror the increase in the level of plastid membrane-bound Put and Spd [3,94]. In fact, ChlTGase was shown to catalyse the binding of ^3^[H]Put and ^3^[H]Spd to photosystem proteins [94]. Substrates of ChlTGases in senescing and non-senescing leaves include apoproteins of the chlorophyll *a*/*b* antenna complex, LHCII, ATP synthase, and pSbS (photosystem II 22kDa protein)—proteins that are essential in energy-dependent quenching and increased thermal dissipation of excessively absorbed light energy in photosystems [7,23,47,55,94]. Several stress-responsive proteins detected in the PA-bound fraction only after induced senescence include the antioxidant enzyme peroxiredoxin, a heat shock protein, ent-copalyl diphosphate synthase, and IAA-amino acid hydrolase [94,95,96,97,98]. The senescence-associated changes in the amount of mono- and bis-(γ-glutamyl)-Put in senescent leaves also corroborated earlier studies on the tobacco corolla. In the latter experimental system, the amount of bis-(γ-glutamyl)-Put and bis-(γ-glutamyl)-Spd decreased and the amount of mono-(γ-glutamyl)-Put increased during petal senescence [50]. In this experimental model, it was also shown that TGase activity was involved in the PCD that takes place following senescence [51]. The fact that PAs, in concert with TGases, are functionally involved in induced leaf senescence is supported by proteomic analyses and TGase activity/transcript modulation [3,94]. The most studied plant gene coding for a protein with TGase activity, *AtPNG1*, is constitutively expressed at low levels in all plant organs during various stages of development and under various light conditions [55]. A similar expression pattern was found for the *HvPNG1*-like homolog in barley. However, *HvPNG1*-like transcription increased as soon as senescence was induced in the dark, concomitant with the start of cell structure disintegration [94]. 

## 5. Conclusions

In this review, we summarise information about plant TGases from studies carried out mainly during the last decade. These enzymes are involved in numerous cellular processes and are present in most plant organs of the species investigated so far. Deeper knowledge would allow us to better understand whether plant TGases are involved in the same basic cellular functions as those of animal TGases. Some features of plant TGases are shared with animal ones, such as their involvement in the wounding response and PCD, as well as in some cellular processes such as cell-to-cell adhesion, which, in plants, occurs in the pollen-style interaction during fertilisation. We also highlight some characteristics that, at least for now, seem to be specific for plant TGases, such as light dependence and apical growth.

In addition to the main biochemical characteristics of plant TGases, we present a bioinformatics analysis of TGases reported from different plant species for the first time. Gene structure results highlight that angiosperms have different intron/exon arrangements than other plant taxa; no substantial differences were observed between monocots and dicots. Furthermore, the bioinformatics analysis allowed us to demonstrate that there are different types of plant TGases, thus supporting the biochemical evidence and the idea that plant TGases are conserved in nature from an evolutionary point of view. 

## Figures and Tables

**Figure 1 cells-11-01529-f001:**
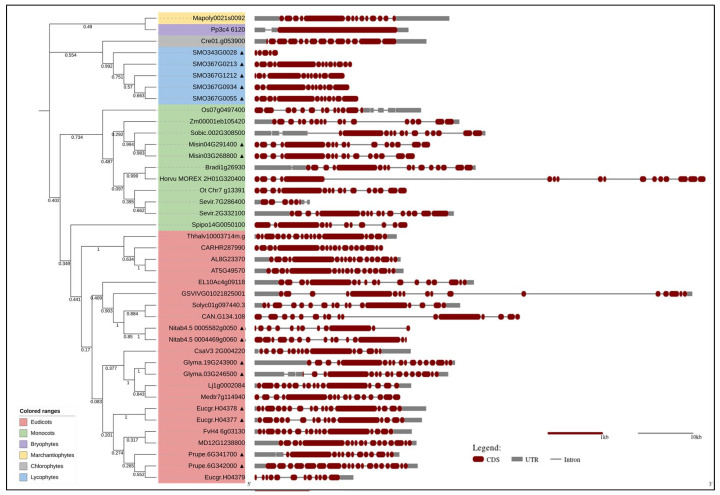
Phylogenetic relationship and gene structure of transglutaminases in various plant species. The phylogenetic analyses were conducted via amino acid sequence alignment using ClustalW and the neighbour-joining method in MEGA-11, and evolutionary distances were computed using the JTT matrix-based method. The proportions of replicate trees in which the associated taxa clustered together in the bootstrap test (1000 replicates) are shown next to the branches. The gene structure, showing the intron/exon pattern, was constructed by The Gene Structure Display Server (GSDS 2.0, (Hu, Jin et al., 2015)) tool (http://gsds.cbi.pku.edu.cn/, accessed on 17 March 2022). Black shaded triangles at the end of gene names represent block/tendem duplication. The transglutaminase genes from various plant species used here were as follows: EL10Ac4g09118 (*Beta vulgaris*); Glyma.03G246500 and Glyma.19G243900 (*Glycine max*); AT5G49570 (*Arabidopsis thaliana*); Nitab4.5_0004469g0060 and Nitab4.5_0005582g0050 (*Nicotiana tabacum*); CsaV3_2G004220 (*Cucumis sativus*); MD12G1238800 (*Malus domestica*); CAN.G134.108 (*Capsicum annuum*); AL8G23370 (*Arabidopsis lyrata*); GSVIVG01021825001 (*Vitis vinifera*); Solyc01g097440.3 (*Solanum lycopersicum*); Eucgr.H04377, Eucgr.H04378, and Eucgr.H04379 (*Eucalyptus grandis*); Medtr7g114940 (*Medicago truncatula*); Lj1g0002084 (*Lotus japonicas*); Prupe.6G341700, Prupe.6G342000 (*Prunus persica*); CARHR287990 (*Cardamine hirsute*); Thhalv10003714m.g (*Eutrema salsugineum*); FvH4_6g03130 (*Fragaria vesca*); Os07g0497400 (*Oryza sativa* ssp. Japonica); Zm00001eb105420 (*Zea mays*); Sobic.002G308500 (*Sorghum bicolor*); Bradi1g26930 (*Brachypodium distachyon*); Misin03G268800 and Misin04G291400 (*Miscanthus sinensis*); Sevir.2G332100 and Sevir.7G286400 (*Setaria viridis*); Spipo14G0050100 (*Spirodela polyrhiza*); Ot_Chr7_g13391 (*Oropetium thomaeum*); Horvu_MOREX_2H01G320400 (*Hordeum vulgare*); Pp3c4_6120 (*Physcomitrium patens*); SMO343G0028, SMO367G0055, SMO367G0213, SMO367G0934, and SMO367G1212 (*Selaginella moellendorffii*); Mapoly0021s0092 (*Marchantia polymorpha*); and Cre01.g053900 (*Chlamydomonas reinhardtii*).

**Figure 2 cells-11-01529-f002:**
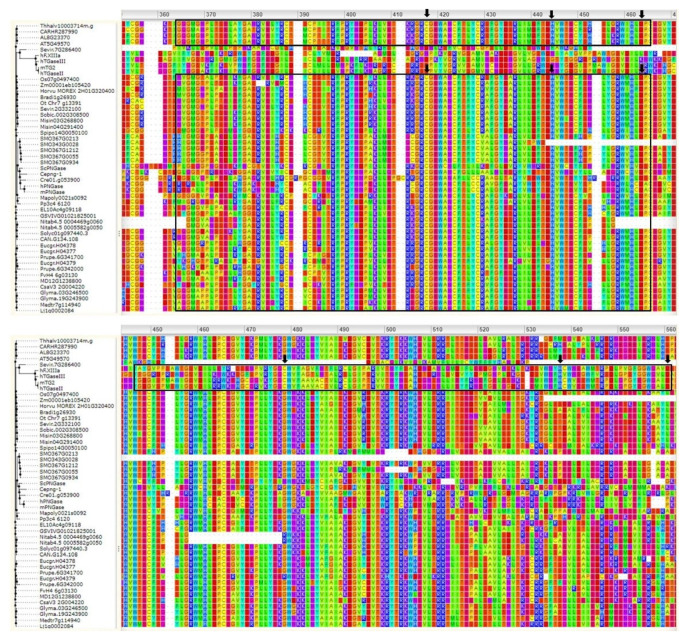
Multiple sequences alignments of transglutaminases and peptide N-glycanases. The protein sequences were aligned using ClustalW, and the ConSurf analysis was performed on the webserver (https://consurf.tau.ac.il/, accessed on 17 March 2022). The sequences in the black boxes show Pfam-PF01841 (Transglut_core). The arrows indicate the Cys–His–Asp amino acids of the catalytic triad. Despite having an amino acid sequence different from those of known animal TGases (Factor XIIIa, hTGaseII, and human TGaseIII), the core domains of the mouse PNGase and that of human TG2 have the same fold, as also reported in the SCOP database (family: transglutaminase core; http://scop.mrc-lmb.cam.ac.uk/scop/data/scop.b.e.d.b.f.html, accessed on 14 March 2022). The protein sequences used here, other than the plants, were: *C. elegans* (Cepng-1, NP_492913.1), *S. cerevisiae* (ScPNGase, NP_015229.1), mouse (mPNGase, NP_067479), human (hPNGase, AF250924.2) and mouse (mTG2, NP_033399.1) peptide N-glycanases, human Factor XIIIa (hF.XIIIa, NP_000120), human TGase II (hTGaseII, NP_945189), and human TGase III (hTGaseIII, Q08188) transglutaminases.

**Figure 3 cells-11-01529-f003:**
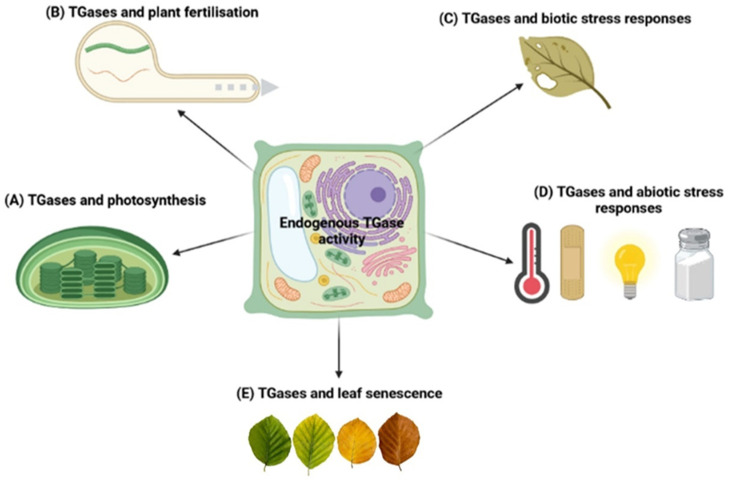
A model summarizing the main physiological roles of plant TGase. TGases have been found in several cellular compartments of different plants. In chloroplasts, TGases contribute to photosynthetic efficiency by increasing the level of bound PAs, ROS scavenging, and CO_2_ assimilation. These aspects are related to the action of TGases on different substrates, e.g., RuBisCO PSII proteins, ATPase, and PMF proteins (**A**). The enzyme is also present in the cytosol, where it can post-translationally modify the cytoskeletal proteins directly or do so via the binding of PAs, contributing to its organisation. This happens in the elongation of the pollen tube (**B**). In plant–pathogen interactions, TGases behave similarly to a PAMP, e.g., the PEP-13 motif of *Phytophthora infestans*. Otherwise, specific plant TGase isoforms might contribute to plant defence mechanisms (**C**). Different abiotic stresses (temperature, wounding, light, and salt stress) stimulate TGase activity, improving plant resilience through the activation of several signalling pathways and the stimulation of several physiological processes, e.g., photosynthetic efficiency, HSP response, antioxidant system, and PA-based cell signalling. By specific inhibitor, TGase inhibition reduces resistance to abiotic stresses and causes decreases in the bound PA contents, decreases in the photochemical efficiency of PSII, and growth reduction (**D**). In the leaf senescence process, TGases increase the accumulation of HSPs and bound PAs. TGases are also involved in the modification of chloroplast proteins and the modulation of anti-senescence enzymes and ATP synthases, finally increasing the photosynthetic efficiency (**E**).

**Table 1 cells-11-01529-t001:** Biochemical features of TGases in higher plants and algae. Plant species, source, molecular weight, optimum pH for activity assay, and localisation in cell compartments are reported; n.d.: not determined.

Plant Species	Source	Molecular Weight (kDa)	Optimum pH Assay	Localisation	References
*Arabidopsis thaliana*	Recombinant enzyme	86	7.5–8.5	Microsomal fraction	[19]
	Entire plant	n.d.	8.4	Entire cell	[20]
*Beta vulgaris*	Leaf	n.d.	7.8	Entire cell	[41]
*Capsicum annuum*	Entire plant	70, 60, 56 64, 34	n.d.	Root	[42]
*Chlamydomonas reinhardtii*	Cell	72	7.4	Cell wall	[27]
*Cucumber sativus*	Cotyledon	77, 58, 50, 30	8.5	Thylakoid	[39]
*Dunaliella salina*	Cell and chloroplast	70, 50, 25	8.5	Chloroplast	[43,44]
*Glycine max*	Leaf and seedling	80	7.6	Entire cell	[45,46]
*Helianthus tuberosus*	Leaf	58, 22, 19	8.0–9.5	Chloroplast	[24]
	Chloroplast	58, 24, 150	n.d.	Thylakoid and stroma	[47]
	Chloroplast	58	8.5	Thylakoid and stroma	[23]
	Etiolated apex	85, 75, 58	8.5	Immature cell	[48]
*Hordeum vulgare*	Leaf	33, 58, 75	8.5	Thylakoid	[3]
*Malus domestica*	Pollen	70, 75	6.5	Chloroplast, microsomal, and cell wall	[49]
*Nicotiana tabacum*	Flower corolla	38, 58	7.5–8.5	Microsome, plastid, and cell wall	[50,51]
	Flower corolla	n.d.	6.5	Epidermis and cell wall	[52]
*Oryza sativa*	Entire plant	40	6.5	Thylakoid	[26]
	Entire plant	72, 39	n.d.	Chloroplast	[36]
*Rosmarinus officinalis*	Entire plant	n.d.	7.0	n.d.	[29]
*Zea mays*	Pollen	58–50; 160–180	n.d.	Pollen	[40]
	Chloroplast	58, 61–67, 77, 150	n.d.	Thylakoid and grana	[28]
	Chloroplast	58	n.d.	Thylakoid	[7]
	Recombinant enzyme	58	8.0	Chloroplast	[14]
	Recombinant enzyme	47	8.0	n.d.	[53]
	Meristematic callus	58, 34	n.d.	Chloroplast and adult leaf	[54]
	Chloroplast	39	8.5	Thylakoid	[19,55]
	Recombinant enzyme	55	8.5	Chloroplast	[56]

## Data Availability

Publicly available datasets were analysed in this study. These data can be found here: https://plants.ensembl.org/index.html (accessed on 14 March 2022).

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
