# Peer review of "Plant Transglutaminases: New Insights in Biochemistry, Genetics, and Physiology"

_cells, 2022, doi:10.3390/cells11091529_

Round 1

Reviewer 1 Report

This review makes an extensive description of the biochemical and bioinformatic characteristics comparing TGases in different plant species. It also adds phylogenetic relationships and genomic organization of TGase proteins. Del Duca et al. devotes an exhaustive section to the physiological role of TGases in plants, focusing on photosynthesis, fertilization, abiotic and biotic stress, and senescence. And finally, it also highlights the close relationship between polyamines and TGases. It seems to me a magnificent review, so I congratulate all the authors.

Author Response

Rev 1.

This review makes an extensive description of the biochemical and bioinformatic characteristics comparing TGases in different plant species. It also adds phylogenetic relationships and genomic organization of TGase proteins. Del Duca et al. devotes an exhaustive section to the physiological role of TGases in plants, focusing on photosynthesis, fertilization, abiotic and biotic stress, and senescence. And finally, it also highlights the close relationship between polyamines and TGases. It seems to me a magnificent review, so I congratulate all the authors.

  1. We are very grateful to the reviewer to appreciate our manuscript.

Reviewer 2 Report

Luigi et al. presented a review on plant transglutaminase. The content of the review is largely based on the works from the corresponding author. Large parts have been covered in the past reviews, especially on the biochemical features, localization, and physiological roles e.g., Serafini-Fracassini and Del Duca (2008), Del Duca et al. (2014), Duarte et al. (2020), etc. The current review is mainly expanding the recent findings of the corresponding author, Del Duca and Serafini-Fracassini with a highlight on the first bioinformatics analysis.

The review needs much improvement as detailed below and highlighted in the attachment to be accepted as a meaningful publication:

  1. The clarity of the review should be improved by clearly stating the take-home messages / insights rather than providing all the details without synthesis.
  2. Summary tables should be comprehensive, e.g., Table 1 should be sorted in a meaningful way with notes on inhibitors and discussed in the text regarding the different optimum pH of the same species. A model or a summary table can be provided to summarize the recent findings on the physiological roles / mechanism of TGases.
  3. Is there a model on the mechanism of TGase in plant physiology, especially in conferring stress tolerance? Please provide one to summarize the review.
  4. Table 2 is unnecessary and can be incorporated as part of Fig. 2, same applied to Fig. 1.
  5. 2 is showing gene structure instead of genome organization. This review never touches on genomics. Please double-check the accuracy based on the latest information, e.g., https://bioinformatics.psb.ugent.be/plaza/versions/plaza_v5_dicots/genes/view/AT5G49570
  6. Please justify the selection in Table 2 from the many plant TGases identified (https://bioinformatics.psb.ugent.be/plaza/versions/plaza_v5_dicots/gene_families/view/ORTHO05D007756).
  7. Fig/Table. 2 should cover all the species listed in Table 1. The authors should make clear if they are defining the different clades of TGases for the first time. If so, please be more comprehensive on the list of selected TGases. Is there only a single gene for each species? an outgroup? Discuss the gene family e.g., gene duplication/expansion/conservation.
  8. Describe/discuss if the phylogenetic tree is consistent with plant evolutionary history. Any differences between dicots and monocots.
  9. How is the conservation of plant TGases compared to that of animals and microbes?
  10. Arrangement in Fig. 3 will be more meaningful if based on the evolutionary relationship.
  11. More comprehensive bioinformatics analysis with clear insight is needed. can perform ConSurf analysis based on Pfam (http://pfam.xfam.org/family/PF01841)
  12. Please elaborate & discuss in detail for TGase activity without the conserved catalytic domains
  13. The authors misused the term homology in place of identity/similarity. Please use the correct terminology to avoid confusion.
  14. Avoid mixing US and UK English. Minor spell check required.
  15. Formatting (Italic for genus and gene name, capitalize, typo, space, inconsistent units)
  16. Please revise the manuscript thoroughly and improve readability, especially rewrite lengthy sentences for clarity. Some sentences are a bit awkward/confusing. 
  17. Please use a proper convention of gene name, e.g., AtPNG1 gene, Atpng1 mutant
  18. I'm not sure whether the content of this review fits "Cellular Biophysics" section.
  19. Many citations are more than 10 years despite the authors' claim that the scope of review is for the past 10 years.
  20. The title should be corrected for grammatical mistakes and revised to fit the content of the review. "Physiological features" doesn't sound right.

Author Response

Rev 2.

Luigi et al. presented a review on plant transglutaminase. The content of the review is largely based on the works from the corresponding author. Large parts have been covered in the past reviews, especially on the biochemical features, localization, and physiological roles e.g., Serafini-Fracassini and Del Duca (2008), Del Duca et al. (2014), Duarte et al. (2020), etc. The current review is mainly expanding the recent findings of the corresponding author, Del Duca and Serafini-Fracassini with a highlight on the first bioinformatics analysis.

The review needs much improvement as detailed below and highlighted in the attachment to be accepted as a meaningful publication.

First of all, we would like to thank you for the time you spent reviewing our Ms. We appreciated the critical review and the constructive suggestions to improve it. The following is the revision according to your comments. In addition, we pay particular attention to single comments and suggestions highlighted in the attached file. Parts, which underwent severe modifications, are highlighted in a yellow font.

  1. The clarity of the review should be improved by clearly stating the take-home messages / insights rather than providing all the details without synthesis.
  2. We guess that in the conclusion section the take home message is clearly reported. Anyway, we revised and rewritten several sentences in the introduction and in the conclusion sections to better clarify the main message of our work.
  3. Summary tables should be comprehensive, e.g., Table 1 should be sorted in a meaningful way with notes on inhibitors and discussed in the text regarding the different optimum pH of the same species. A model or a summary table can be provided to summarize the recent findings on the physiological roles / mechanism of TGases.
  4. Is there a model on the mechanism of TGase in plant physiology, especially in conferring stress tolerance? Please provide one to summarize the review.
  5. Thanks to the reviewer for suggestions. We revised Table 1 in light to improve the readability and shortened it in a meaningful way. Please see the new version in the text. We also added a model figure (Figure 3) to summarize the physiological roles of plant TGase.
  6. Table 2 is unnecessary and can be incorporated as part of Fig. 2, same applied to Fig. 1.
  7. The bioinformatic section has been modified according to the reviewer’s comments. Please see the new version of the text and figures.
  8. Fig. 2 is showing gene structure instead of genome organization. This review never touches on genomics. Please double-check the accuracy based on the latest information, e.g., https://bioinformatics.psb.ugent.be/plaza/versions/plaza_v5_dicots/genes/view/AT5G49570
  9. Please see the above response.
  10. Please justify the selection in Table 2 from the many plant TGases identified (https://bioinformatics.psb.ugent.be/plaza/versions/plaza_v5_dicots/gene_families/view/ORTHO05D007756).
  11. The model plant species, described by Chang et al., (Cell. 2016; 167(2):325-339), available in the Plaza database (https://bioinformatics.psb.ugent.be/plaza/) were selected for this analysis.
  12. Fig/Table. 2 should cover all the species listed in Table 1. The authors should make clear if they are defining the different clades of TGases for the first time. If so, please be more comprehensive on the list of selected TGases. Is there only a single gene for each species? an outgroup? Discuss the gene family e.g., gene duplication/expansion/conservation.
  13. The plant species Medicago sativa, Helianthus tuberosus, Dunaliella salina and Rosmarinus officinalis in Table 1 are not available in the database, and not covered in bioinformatics analysis. The other corrections have been made in the Ms as per the reviewer’s suggestion.
  14. Describe/discuss if the phylogenetic tree is consistent with plant evolutionary history. Any differences between dicots and monocots.
  15. Description of phylogenetic tree in context with plant evolutionary history has been included in the manuscript as suggested by the reviewer.
  16. How is the conservation of plant TGases compared to that of animals and microbes?
  17. The conservation of plant TGases compared to that of animals and microbes has been included in the manuscript as suggested by the reviewer.
  18. Arrangement in Fig. 3 will be more meaningful if based on the evolutionary relationship.
  19. According to other suggestions of the referee, the entire bioinformatic section has been revised and rewritten. Please see the new version of Ms.
  20. More comprehensive bioinformatics analysis with clear insight is needed. can perform ConSurf analysis based on Pfam (http://pfam.xfam.org/family/PF01841)
  21. Thanks to review for the suggestion. The ConSurf analysis has been performed and included in the manuscript as suggested by the reviewer.
  22. Please elaborate & discuss in detail for TGase activity without the conserved catalytic domains
  23. The discussion about the TGase activity without catalytic domain has been incorporated in theMs, as suggested by the reviewer.
  24. The authors misused the term homology in place of identity/similarity. Please use the correct terminology to avoid confusion.
  25. Thanks to the reviewer for accurate revision of our Ms. The entire text has been modified accordingly.
  26. Avoid mixing US and UK English. Minor spell check required.
  27. We are grateful to the reviewer for the particular attention to our paper. We checked the whole Ms and added details whenever required; we paid specific attention to US/UK English. The language has been revised in the whole MS, paying specific attention to the abovementioned suggestions.
  28. Formatting (Italic for genus and gene name, capitalize, typo, space, inconsistent units)
  29. We checked the whole Ms and changed details whenever required. The whole MS has been revised.
  30. Please revise the manuscript thoroughly and improve readability, especially rewrite lengthy sentences for clarity. Some sentences are a bit awkward/confusing. 
  31. Thank you to the referee for this criticism, the text has been modified accordingly. Specific attention was spent to the abovementioned suggestions. In particular, tempi, prepositions, abbreviations, singular/plural use, etc. have been checked. Moreover, several sentences have been rewritten.
  32. Please use a proper convention of gene name, e.g., AtPNG1 gene, Atpng1 mutant
  33. We are grateful to the reviewer for the suggestion. We checked and modified it accordingly with the reviewer’s suggestion.
  34. I'm not sure whether the content of this review fits "Cellular Biophysics" section.
  35. We received a formal invitation to participate in the special issue (Function of Transglutaminases in Adhesion Dynamics, Differentiation, and Cell Survival) from the Guest Editor, so this issue depends on Guest Editor/Journal. Also, we think that the important functions of the TGase enzyme in plants is relevant for readers of Cells journal.
  36. Many citations are more than 10 years despite the authors' claim that the scope of review is for the past 10 years.
  37. As reported in the title of the article "Plant Transglutaminase: new and old insights of biochemical, genetic and physiological traits" it has been necessary to add some basic knowledge from which to discuss about the literature of the last 10 years. Therefore, most of the cited papers are from the last 10 years (but some important information derives from previous papers).
  38. The title should be corrected for grammatical mistakes and revised to fit the content of the review. "Physiological features" doesn't sound right.
  39. Thanks to the reviewer for the suggestion. The title has been revised and corrected. The new title is: “Plant Transglutaminase: new and old insights of biochemical, genetic and physiological traits”
  40. Pep-13, is this unique to pathogen/microbe?
  41. To date in literature, Pep-13 was reported as the only pathogen/microbe motif in the TGase sequence.

Round 2

Reviewer 2 Report

The revised manuscript is poorly formatted without line numbers and the changes according to reviewers' comments are not clearly described in the response. It's not clear what has been improved in the manuscript according to different reviewers.

The new title remains grammatically wrong. A suggestion: "Plant Transglutaminase: new and old insights in biochemistry, genetics, and physiology"

The authors still use the misleading term "genome organisation" to refer to "gene structure" in the manuscript.

Check again Table 1 for accuracy in singular/plural & formatting. Pichia pastoris transformation doesn't fit into localisation. Specify what's n.d. & revise table caption. Many errors highlighted in the previous attachment are still present in the revision.

The manuscript needs a professional English editing service to improve readability as there are still many inconsistencies & grammatical mistakes.

Author Response

Cover Letter

            Bologna, April 26th 2022

Dear Editor and Dear Referee,

We revised the Ms accordingly to suggestions and criticisms of reviewer 2. In the new file, we highlighted in yellow font all the changes made. Please see the new version of Ms.

We have the pleasure to submit our revised version of review for Cells, Special Issue on ‘Transglutaminase 2 (TG2) in Cell Adhesion Dynamics, Cell Differentiation and Cell Survival’, in agreement with the kind invitation of guest editors. The article is now entitled: “Plant transglutaminases: new insights in biochemistry, genetics, and physiology” (cells-1643834).

The manuscript has been revised accord to referee’s criticisms as reported in the responses to reviewer.

We confirm that neither the manuscript nor any parts of its content are currently under consideration or published in another journal. All authors have approved the manuscript and agree with its submission to Cells.

With kind regards,

Stefano Del Duca

Full Professor of Botany,

Dept. of Biological, Geological and Environmental Sciences (BiGeA),

University of Bologna, via Irnerio 42,

40126 Bologna,

Responses to referee criticisms.

  1. The revised manuscript is poorly formatted without line numbers and the changes according to reviewers' comments are not clearly described in the response. It's not clear what has been improved in the manuscript according to different reviewers.
  2. The line numbers were added to the revised Ms. All the changes in Ms were highlighted in yellow and made according to the criticisms of reviewer 2.
  3. The new title remains grammatically wrong. A suggestion: "Plant Transglutaminase: new and old insights in biochemistry, genetics, and physiology"
  4. Thanks to the referee for the criticism. The title was corrected as follows: “Plant transglutaminases: new insights in biochemistry, genetics, and physiology”.
  5. The authors still use the misleading term "genome organisation" to refer to "gene structure" in the manuscript.
  6. We checked and corrected the right terminology in the entire Ms.
  7. Check again Table 1 for accuracy in singular/plural & formatting. Pichia pastoris transformation doesn't fit into localisation. Specify what's n.d. & revise table caption. Many errors highlighted in the previous attachment are still present in the revision.
  8. Thanks to reviewer for specific attention to our Ms. We revised the table 1, modifying “Pichia pastoris transformation” to “n.d.”. In addition, we added in table caption the meaning of n.d. (n.d., not determined). Errors still present in the Ms were checked and corrected, other errors found was corrected.
  9. The manuscript needs a professional English editing service to improve readability as there are still many inconsistencies & grammatical mistakes.
  10. An English mother tongue speaker who has strong expertise in plant biology revised the language in the whole MS. Specific attention was spent to the abovementioned suggestions. In particular, tempi, prepositions, abbreviations, singular/plural use, etc. have been checked.
